

# Assessing the Effectiveness of SO₂, NOx, and NH₃ Emission Reductions in Mitigating Winter PM2.5 in Taiwan Using CMAQ Model

Ping-Chieh Huang[1], Hui-Ming Hung[1*], Hsin-Chih Lai[2], and Charles C.-K. Chou[3]

[1]Department of Atmospheric Sciences, National Taiwan University, Taipei, 106319, Taiwan
[2]Department of Occupational Safety and Health, Chang Jung Christian University, Tainan, 711301, Taiwan
[3]Research Center of Environmental Changes, Academia Sinica, Taipei, 115201, Taiwan

*Correspondence to*: Hui-Ming Hung (hmhung@ntu.edu.tw)

**Abstract.** Taiwan experiences higher air pollution in winter when particulate matter (PM2.5) levels frequently surpass national standards. This study employs the Community Multiscale Air Quality model to assess the effectiveness of reducing NH₃, NOx, and SO₂ emissions on PM2.5 secondary inorganic species (i.e., $SO_4^{2-}$, $NO_3^-$, and $NH_4^+$). For sulfate, ~ 43.7% is derived from the chemical reactions of local SO₂ emission, emphasizing the substantial contribution of regional transported sulfate. In contrast, local NOx and NH₃ emissions predominantly influence nitrate and ammonium. Reducing SO₂ emissions decreases sulfate, thereby influencing NH₃ partitioning and resulting in a decreased ammonium. Similarly, reducing NOx emissions lowers HNO₃, impacting nitrate and ammonium concentrations due to changes in HNO₃ and NH₃ partitioning. A particularly significant finding is that NH₃ emissions reduction decreases not only nitrate and ammonium but also sulfate by altering cloud droplet pH and SO₂ oxidation processes. While SO₂ reduction's PM2.5 impact is less than NOx and NH₃, it emphasizes the complexity of regional sensitivities. Most of western Taiwan is NOx-sensitive, so reducing NOx emissions has a more substantial impact on lowering PM2.5. However, given the higher mass emissions of NOx than NH₃ in Taiwan, NH₃ has a more significant consequence in mitigating PM2.5 per unit mass emission reduction. The cost-effectiveness analysis suggests that NH₃ reduction outperforms SO₂ and NOx. Nevertheless, cost estimates vary due to methodological differences and regional emission sources. Overall, this study considers both efficiency and costs, highlighting NH₃ emissions reduction as a promising strategy for PM2.5 mitigation in the studied Taiwan's environment.



## 1 Introduction

Aerosol particles in the atmosphere have become a significant concern due to their adverse health effects (Maynard et al., 2002; Shiraiwa et al., 2017; Sugiyama et al., 2020) and their role in affecting global radiation budgets (Ramanathan et al., 2001; IPCC, 2021). Long-term exposure to air pollutants such as particulate matter (PM) and ozone ($O_3$) has been linked to

millions of premature deaths annually on a global scale (Vohra et al., 2022; WHO, 2021). These findings emphasize the critical need for a comprehensive understanding of air pollution and effective management strategies to protect public health and mitigate environmental consequences.

PM can enter the atmosphere through direct emissions (primary aerosols), such as black carbon, sea salt, dust, and certain organic substances, or it can be formed via chemical reactions involving gas-phase precursors (secondary aerosols), such as

sulfate ($SO_4^{2-}$), nitrate ($NO_3^-$), and ammonium ($NH_4^+$) (Seinfeld et al., 2006). The composition of PM varies globally, with organic and inorganic components representing major categories. Inorganic aerosol components, including sulfate, nitrate, ammonium, and chloride, constitute 35 % to 77 % of $PM_1$ worldwide (Schroder et al., 2018). The significant proportion of secondary inorganic composition can influence the pH value of PM, further impacting the formation of secondary organic matter (Zhang et al., 2007).

Sulfate is formed through both gas phase and aqueous oxidation of sulfur dioxide ($SO_2$) emitted from sources like coal power plants and industrial processes, while nitrate is produced via the oxidation of nitrogen oxides (NOx), mainly emitted from traffic. Ammonium can be formed through the partitioning between aqueous and gas phases of ammonia ($NH_3$) emitted from agricultural and industry sources. The overall sulfate-nitrate-ammonium formation processes are illustrated in Fig. 1. In addition to the gas phase reaction with OH radicals, $SO_2$ can also be oxidized, such as hydrogen peroxide ($H_2O_2$) or ozone ($O_3$)

in the aqueous phase. Due to the low volatility and high dissociation constant of sulfuric acid, most sulfuric acid is in the condensed phase and dissociates in aqueous particles. Ammonia and nitric acid are semi-volatile, so their dissolution in particles is determined by their Henry's law constant and dissociation constants. The presence of acid for ammonia or base for nitric acid can promote individual dissolution (Seinfeld et al., 2006). The interaction of sulfate, nitrate, and ammonium is vital in determining the quantity of PM.

Human activities and natural sources are responsible for releasing the inorganic aerosol precursors, i.e., $SO_2$, NOx, and $NH_3$. Reducing these emissions might mitigate $PM_{2.5}$ levels, thus improving air quality. Numerous studies have investigated emission reduction strategies, with a focus on $NH_3$ reductions showing promise in decreasing $PM_{2.5}$ levels (Chen et al., 2019; Gu et al., 2021). Liu et al. (2019) used the WRF-chem model to investigate emission reduction strategies in China and found that reducing $SO_2$ and NOx emissions alone does not significantly reduce total $PM_{2.5}$ levels. However, including controls for

$NH_3$ emissions can reduce $PM_{2.5}$ by approximately 11-17 %, but with the potential risk of exacerbating acid rain. Derwent et al. (2009) employed a photochemical trajectory model to simulate PM concentrations in the UK with 30 % reductions in $NH_3$, $SO_2$, NOx, VOC, and CO emissions. In an ammonium-limited environment (southern UK), $NH_3$ emissions reductions had the most significant impact on PM reduction, exhibiting a non-linear dynamic effect.



In Taiwan, secondary inorganic aerosol constitutes 30-53 % of $PM_{2.5}$, with sulfate, nitrate, and ammonium contributing
significantly (16-32 %, 2-24 %, and 6-12 %, respectively) (Chuang et al., 2021). $PM_{2.5}$ concentrations in Taiwan are usually
higher in winter than in summer due to the influence of meteorological conditions and the planetary boundary layer height.
Especially on the leeward side of the prevailing northeast monsoon in winter (i.e., the western Taiwan), the impact of $PM_{2.5}$
concentration accumulation is more significant (Hsieh et al., 2022; Hsu et al., 2016; Lai et al., 2020). Even though the $PM_{2.5}$
concentration has decreased over the past two decades (Cheng et al., 2019; Chuang et al., 2021), the current $PM_{2.5}$ reduction
effort might not efficiently meet the standard set by the Taiwan Ministry of Environment (TW-MOENV): a 24-hour standard
of 35 μg m$^{-3}$ and annual level of 15 μg m$^{-3}$. Due to the complex interactions among secondary inorganic components and
their substantial contribution to total PM in Taiwan, further research is imperative.

To study air pollution in Taiwan, we employed the Community Multiscale Air Quality (CMAQ) model, which is recognized
for its comprehensiveness in simulating atmospheric chemical processes. CMAQ model incorporates various chemical
processes, including photolysis, multiphase chemistry, aerosol microphysics, aqueous chemistry in clouds, and cloud
formation on particles (Byun et al., 2006). It is widely used to assess air pollutants on a regional scale and helps understand
changes and mechanisms of pollutants under different scenarios. This study focuses on investigating the formation of
secondary inorganic species, specifically sulfate, nitrate, and ammonium, during winter in Taiwan. With an understanding
of the contribution of each composition contribution from different processes and their interaction, the reduction efficiency
and cost of each aerosol precursor (i.e., $SO_2$, NOx, and $NH_3$) in mitigating $PM_{2.5}$ are evaluated.

## 2 Methodology

### 2.1 CMAQ model

The Community Multiscale Air Quality (CMAQ) model with the Weather Research and Forecasting (WRF) model for
meteorological conditions was applied to simulate the concentrations of various chemical species over Taiwan. The WRF
model version 3.7.1 (Skamarock et al., 2008) was initialized using a cold start and simulated the period from 28$^{th}$ November
2018 to 31$^{st}$ December 2018, with analysis focusing on December only. Four nested domains, as shown in Fig. S1a, were
creased with horizontal resolutions of 81, 27, 9, and 3 km and a total of 45 vertical layers. The outer domain covers most of
East Asia and the western Pacific, while the inner domain was dedicated to Taiwan.

The CMAQ model version 5.2.1 (Byun et al., 2006; Wyat Appel et al., 2018) was set up using the same horizontal grid
structure as WRF, but with 15 vertical layers as seven layers under 1500 m and the top layer ∼17 km above the ground. The
inner domain of CMAQ consists of $135 \times 90$ grid cells. The chemical mechanism used in the simulations was Carbon Bond
Mechanism version 6 and aerosol module version 6 with aqueous chemistry (cb6r3_ae6_aq). Emission data for Taiwan were
from the Taiwan Emission Data System (TEDS9.0) based on the 2013 data. TEDS9.0 provides comprehensive information
on various sources of pollutants in Taiwan, including industrial processes, transportation, energy production, and residential



activities. Hourly model output data enables detailed temporal analysis. Additional details on the model configuration, including physical and chemical mechanisms, are summarized in Tables S1 and S2.

## 2.2 Observation data in Taiwan

The simulated data of the control run were compared with observations from ground-based monitoring stations to validate the model outputs. Hourly meteorological parameters (air temperature, relative humidity, and wind field) and pollutants (CO, O$_3$,

and PM$_{2.5}$) data were collected from TW-MOENV air quality monitoring stations. Four stations along the western coast of Taiwan (Fig. S1b) - Tamsui (25.16° N, 121.45° E), Shalu (24.23° N, 120.57° E), Taixi (23.72° N, 102.20° E), and Qianzhen (22.61° N, 120.31° E), were selected for wind fields and PM$_{2.5}$ concentrations comparisons. Additionally, intensive observation data from Shalu (24.24° N, 120.57° E), Chung Shan Medical University (24.12° N, 120.65° E, CSMU), Zhushan (23.76° N, 120.68° E), and Xitou (23.67° N, 120.80° E) in central Taiwan from 1$^{st}$ December to 21$^{st}$ December 2018, provided further

inside into PM$_{2.5}$ and its components. Sampling occurs from 9:00 to 18:00 for daytime samples and from 21:00 to 6:00 (next day) for nighttime samples. Inorganic ions were analyzed using ion chromatography (IC). More details of the analytical methodology can be found in Chen et al. (2021) and Lee et al. (2019).

## 2.3 CMAQ experimental design

To evaluate the contribution of the sulfate pathway and the impact of aerosol precursor emission reduction at mitigating PM$_{2.5}$

levels, we designed the following two series of experiments.

### 2.3.1 Sulfate contribution

The local sulfate in PM$_{2.5}$ (PM-sulfate) can be contributed from transport and local gas phase and aqueous phase chemical reactions. To assess the contribution of each source to the local PM-sulfate within the inner domain, adjustments were made to the chemical reaction module within the CMAQ chemistry-transport model (CCTM). This analysis involved two

simulations: "NoAqChem run" and "NoChem run". In NoAqChem run, sulfur aqueous phase oxidation reactions, including S(IV) oxidation by O$_3$, H$_2$O$_2$, organic peroxides, and metal ion catalysis (Jacobson, 1997), were turned off. In NoChem run, all chemical reactions in CMAQ were disabled. By comparing the PM-sulfate of these simulations with the control run, the contribution fractions of gas phase ($F_{gas}$) and aqueous phase ($F_{aq}$) reactions to local PM-sulfate were evaluated using the following equations:

$$F_{gas} = \frac{NoAqChem\ run - NoChem\ run}{Control\ run} \times 100\ \% \tag{1}$$

$$F_{aq} = \frac{Control\ run - NoAqChem\ run}{Control\ run} \times 100\% \ . \tag{2}$$





### 2.3.2 Emission reduction efficiency

Our study assessed variations in PM$_{2.5}$ and major inorganic composition concentrations resulting from emissions reductions. Specifically, we focused on modifying SO$_2$, NOx, and NH$_3$ emissions proportionally, key aerosol precursors forming sulfate,

nitrate, and ammonium in aerosols. The emissions of SO$_2$, NOx, and NH$_3$ are $1.18 \times 10^6$, $4.61 \times 10^6$, and $1.77 \times 10^6$ tons yr$^{-1}$, respectively, for applied emission inventory. Emissions were reduced individually at intervals of 0.2 (i.e., 0.8x, 0.6x, 0.4x, and 0.2x of the control-run emissions) in the inner domain, labeled as the "ER1 runs". Additionally, the effects of potential earlier emission quantities were explored by increasing emissions at 0.5 intervals (i.e., at 1.5x and 2.0x of the control-run emissions), referred to as "EI runs." Considering the interplay between nitrate and ammonium due to acid-base balance, we conducted

"ER2 runs", reducing both NOx and NH$_3$ emissions at 0.2 intervals. Notably, ER2 runs cover the first half of December (from 1$^{st}$ December to 14$^{th}$ December) to save computing resources, while other simulations encompassed the entire month. Table 1 provides a summary of all simulation settings.

The variation of aerosol composition and PM quantity based on emission adjustment is evaluated to assess the sensitivity and effectiveness of emission reduction. Following Takahama et al. (2004), a dimensionless sensitivity coefficient ($S_{X,Y}$) was

introduced to evaluate the potential impacts of $X$ emission reduction on $Y$ (nitrate or PM$_{2.5}$) as follows:

$$S_{X,Y} = \frac{E_X}{Y}\frac{dY}{dE_X} = \frac{d\log(Y)}{d\log(E_X)} \approx \frac{\Delta\log(Y)}{\Delta\log(E_X)}, \tag{3}$$

where $E_X$ represents the specific emission of SO$_2$, NOx, and NH$_3$. $\Delta(var)$ is the difference between $var_i$ and $var_{i-1}$, two adjacent points. For $Y =$ nitrate, the sensitivity is sensitive to NOx and NH$_3$. A higher response among the given emission reductions indicated the properties of the environment, such as NOx-sensitive or NH$_3$-sensitive (Petetin et al., 2016). This framework can

also assess the potential sensitivity of emission reductions on PM$_{2.5}$ concentration (i.e., $Y =$ PM$_{2.5}$) for each emission. A higher sensitivity under $E_X$ reduction indicates that more significant PM$_{2.5}$ mitigation can be achieved by controlling this emission.

### 2.4 Box model

A simplified box model constructed using Python was developed to study the influence of NH$_3$ emissions on sulfate formation, specifically focusing on the chemical reactions occurring in the aqueous phase, including dissolution, oxidation, and

dissociation processes (Reactions 5-8 in Table 2). The model aimed to assess the impact of ammonia emission reduction on sulfate formation, focusing on chemical processes only with fixed meteorological conditions and no physical transport. The box model conditions were adapted from a grid point of CMAQ within the planetary boundary layer exhibiting sufficient liquid water content (LWC). To retrieve the initial concentrations of reactants, the maximum attention of oxidants (O$_3$ and H$_2$O$_2$) along the ammonia reduction profile was applied, with an equal amount of sulfate back to SO$_2$. The input parameters from

CMAQ, including air temperature (T), liquid water content, and concentrations of SO$_2$, carbon dioxide (CO$_2$), total nitric acid (HNO$_3$), total NH$_3$, H$_2$O$_2$, O$_3$, S(IV), iron (Fe(III)), and manganese (Mn(II)). A summary of the initial conditions employed in this study is provided in Table S3. Similar to the aqueous phase reaction of CMAQ, the dissolution of chemical components



in water follows the equilibrium between the gas-to-dissolved phase controlled by Henry's constants. The initial pH value was calculated based on the acid-base balance and charge balance Eq. (4) of the system, ensuring consideration among different chemical species and their influence on the overall pH.

$$[\text{H}^+] = [\text{OH}^-] + [\text{HCO}_3^-] + 2[\text{CO}_3^{2-}] + [\text{HSO}_3^-] + 2[\text{SO}_3^{2-}] + 2[\text{SO}_4^{2-}] + [\text{NO}_3^-] - [\text{NH}_4^+] \tag{4}$$

At each time step, the model calculated concentration changes following the oxidation reactions, and the pH value was recalculated at the new equilibrium state. The oxidation reactions considered in the box model are as follows (Seinfeld et al., 2006):

$$\text{SO}_2 + \text{O}_3 + \text{H}_2\text{O} \rightarrow \text{SO}_4^{2-} + \text{O}_2 + 2\text{H}^+ \tag{R1}$$

$$\text{HSO}_3^- + \text{O}_3 \rightarrow \text{SO}_4^{2-} + \text{O}_2 + \text{H}^+ \tag{R2}$$

$$\text{SO}_3^{2-} + \text{O}_3 \rightarrow \text{SO}_4^{2-} + \text{O}_2 \tag{R3}$$

$$\text{HSO}_3^- + \text{H}_2\text{O}_2 + \text{H}^+ \rightarrow \text{SO}_4^{2-} + 2\text{H}^+ + \text{H}_2\text{O} \tag{R4}$$

$$\text{S(IV)} + \frac{1}{2}\text{O}_2 \xrightarrow{\text{Mn}^{2+},\text{Fe}^{3+}} \text{S(VI)}, \tag{R5}$$

with the rate constants summarized in Table S4.

Two sets of experiments were conducted to compare the results with those of the CMAQ. These simulations were conducted by gradually reducing $NH_3$ emissions at 0.1x intervals. The first set exclusively considered the oxidation reactions of S(IV) by $O_3$ and $H_2O_2$ (i.e., R2+R3), while the second set incorporated additional oxidation reactions of S(IV) by $O_2$, with catalysis of iron and manganese (i.e., R2+R3+R4). The timestep of these experiments was 0.05 seconds, and results from a 10-minute run were analyzed as the oxidation reaction levels ranged from 65.1 to 99.9 % compared to the 1-hour reaction, depending on the emission reduction ratio. The box model results reflect the impact of ammonia emission on sulfate formation under specific conditions. However, the composition of a grid box in CMAQ is influenced by chemical processes and transportation. The overall results between the box model and CMAQ may not match precisely.

## 2.5 Mitigation efficiency and cost estimation

To evaluate the effectiveness of $PM_{2.5}$ reduction, we employed an exponential function to fit $PM_{2.5}$ concentration as a function of emission adjustment ratios ranging from 0.2x to 2.0x of control-run emissions. The derivative of $PM_{2.5}$ or the quantity of a specific component ($Y$) concerning emissions was applied to assess the emission reduction efficiency of $X$ ($X$ can be $SO_2$, NOx, or $NH_3$), denoted as follows:

$$Y \ reduction \ efficiency = \frac{dY}{dE_X} \ [\mu\text{g m}^{-3} \cdot \text{ton}^{-1}\text{yr}] \tag{5}$$

The cost of $PM_{2.5}$ reduction is evaluated by dividing the marginal abatement cost (MAC) by $PM_{2.5}$ reduction efficiency (obtained from Eq. (5) with $y$ as $PM_{2.5}$) as follows:

$$PM_{2.5} \ reduction \ cost = \frac{MAC}{PM_{2.5} \ reduction \ efficiency} \ [\text{USD yr}^{-1} \cdot \mu\text{g}^{-1}\text{m}^3]. \tag{6}$$



The applied MAC values are 421-1630 USD ton$^{-1}$, 8152-9570 USD ton$^{-1}$, and 1318-1400 USD ton$^{-1}$ for SO$_2$, NOx, and NH$_3$, respectively, based on the studies of Gu et al. (2021) and Kaminski (2003).

## 3 Results and Discussion

### 3.1 Model performance

#### 3.1.1 Meteorology

The comparison between WRF model results and TW-MOENV observations is presented in Tables 3 and S5, providing a comprehensive overview of monthly mean values, correlation coefficients (r), mean bias errors, mean absolute error, mean fractional bias, and mean fractional errors. Notably, the correlation coefficients for air temperature consistently exceed 0.8 across all four stations, showcasing a robust agreement. For relative humidity, the correlation coefficients range from 0.71 to 0.86, indicating a good alignment between observation and model results. For wind speed, the correlation coefficients range from 0.42 to 0.85 at these stations. The mean bias error at Shalu and Qianzhen meets the criteria suggested by Hu et al. (2016), while the mean absolute error at Tamsui, Shalu, and Qianzhen meets the criteria. Taixi is mostly underestimated in the model to have a higher mean absolute error. Overall, these findings demonstrate a satisfactory performance of the model.

Wind fields play a critical role in the dispersion of air pollutants, affecting their transport and spatial distribution, not only for wind speed but for wind direction. Fig. S2 illustrates that the model reasonably captures the prevailing winter wind patterns, characterized by predominant winds blowing from the northeast. Although discrepancies in wind speed exist, with slight underestimations in Taixi and overestimations in Tamsui, Shalu, and Qianzhen, the overall trend of strong and weak winds is consistent between the model and observations.

#### 3.1.2 Air pollutants

Table S5 also provides statistical results for pollutants. The correlation coefficients range from 0.46 to 0.62 for CO and from 0.58 to 0.84 for O$_3$. The mean bias errors are higher for both CO and O$_3$, likely due to a significant underestimation of CO and an overestimation of O$_3$ in the model results. For PM$_{2.5}$, the model exhibits good agreement with observations, capturing similar concentration patterns. Specifically, lower PM$_{2.5}$ concentrations were observed under more vital northeasterly wind conditions, while weaker northeasterly winds or winds from other directions corresponded to higher pollutant concentrations. The correlation coefficients for PM$_{2.5}$ concentration range from 0.42 to 0.71, and the mean fractional bias and mean fractional error for PM$_{2.5}$ are within the criteria (Table 3), affirming the model's reliability (Fig. S2).

For the spatial distribution of PM$_{2.5}$, areas with high pollution levels are primarily concentrated in western regions, corresponding to densely populated areas (Fig. S3a). The PM$_{2.5}$ concentration gradually increases from north to south, mainly over flat land areas. To assess the regional distribution, we used regional average concentration and partitioning of PM$_{2.5}$,



based on TW-MOENV's pollutant zone classification (Fig. S3b), with altitudes less than 200 m to avoid complexities in terrain. The partitioning of $PM_{2.5}$ is similar across regions, with secondary inorganic components constituting more than half of $PM_{2.5}$. This study focuses on central Taiwan, specifically the marked red area on the map. In the control run, the surface layer mean

$PM_{2.5}$ in central Taiwan has a similar pattern to nitrate and ammonium, while sulfate has some slight differences (blue line in Fig. S4). The correlation coefficient between $PM_{2.5}$ and sulfate, nitrate, and ammonium are 0.65, 0.96, and 0.95, respectively. Given the high correlation between nitrate and ammonium (r = 0.98) and the significant contribution of nitrate to $PM_{2.5}$ concentrations in Taiwan, nitrate emerges as a major contributor to $PM_{2.5}$. In addition, we notice that the concentration of pollutants is related to the strength of the wind field. Combined with Shalu's wind field time series diagram, representing the

environmental wind in central Taiwan, nitrate, ammonium, $PM_{2.5}$, and wind speed have a certain negative correlation, while sulfate is less relevant. This suggests that gaseous $HNO_3$ and $NH_3$ accumulate locally during weak wind conditions, facilitating the transformation of nitrate and ammonium into aerosol particles.

The correlation coefficients of $PM_{2.5}$ of Shalu and CSMU are 0.76 and 0.65, respectively, and the consistency of concentration and change trend in these two stations can also be seen in Fig. S5. However, the correlation between observational data and

model data at Zhushan and Xitou is poor, which may be due to the influence of the complex topography of these two places. The simulated proportions of PM-sulfate, PM-nitrate, and PM-ammonium formation by CMAQ are 9.1-11.4 %, 18.7-34.9 %, and 9-13.7 %, respectively. In contrast, the observation data indicates that proportions of PM-sulfate, PM-nitrate, and PM-ammonium formation are 13.9-19.6 %, 16.6-22.8 %, and 7.6-10.7 %, respectively (Fig. 2). Considering the spatial heterogeneity of $PM_{2.5}$, our analysis mainly discusses on examining the composition of $PM_{2.5}$ rather than emphasizing the

differences between the model outputs and observational data. Overall, in central Taiwan's average model data and single-point observation data, secondary inorganic aerosols account for approximately half of the concentration of $PM_{2.5}$, of which nitrate is the highest.

### 3.2 Sulfate formation pathway on $PM_{2.5}$

With the analysis of NoAqChem and NoChem runs, the mean contributions of sulfate in central Taiwan are as follows: 13.2

% from gas reactions, 30.5 % from aqueous reactions, and 56.3 % from other processes, including the transportation from domain boundary, locally emitted primary sulfate (constituting less than 5 % of $SO_2$ emission), and alterations of deposition. The analysis for other areas is summarized in Table S6, with all chemical processes having a portion less than 50 %. The major aqueous reactions occur in the cloud, typically with higher cloud water content (QC). By comparing the time series of average cloud water content in the boundary layer (Fig. S6a), it becomes evident that high QC corresponds to dominant aqueous

chemical processes in sulfate formation. The correlation coefficient between QC and the sulfate difference (control run - NoAqChem run) is 0.65 (indicating the contribution of aqueous phase chemical processes). More details regarding the correlation coefficients of PM concentration and meteorological parameters are available in Table 4. Nitrate and ammonium concentrations exhibit a stronger relationship with the wind field, while sulfate concentration is more influenced by the





occurrence of aqueous phase chemistry, specifically the amount of cloud water content in the atmosphere. In addition, the
contribution of aqueous chemical processes is also highly correlated with sulfate concentration in the control run, particularly
during periods of elevated sulfate levels. However, the impact of these sulfate changes on nitrate, ammonium, and $PM_{2.5}$ is
insignificant (Fig. S4). Given that nitrate comprises a significant proportion of PM during the winter in Taiwan, our results
suggest that the overall trend of total $PM_{2.5}$ aligns more closely with the wind field.

### 3.3 Emission affecting $PM_{2.5}$ on the surface layer

### 3.3.1 Trends of concentration

The impact of emission adjustments on $PM_{2.5}$ and its components in central Taiwan is shown in Fig. 3. $PM_{2.5}$ and secondary
inorganic components show a decreasing trend as the emission ratio is reduced. At emission ratio larger than 1, $PM_{2.5}$ variation
is relatively flat compared with emission ratio less than 1. This indicates a higher $PM_{2.5}$ mitigation efficiency for future
emission reduction. Reductions of $SO_2$ primarily decrease sulfate and ammonium, while NOx reductions affect nitrate and
ammonium. However, $NH_3$ reductions decrease ammonium, nitrate, and sulfate. Since $SO_2$ is a sulfate precursor, reducing $SO_2$
decreases sulfate formation, consequently modifying the ammonia partition and decreasing ammonium (Tsimpidi et al., 2007).
The negligible impact on nitrate can be attributed to nitric acid partition processes, affected by particle acidity and available
aerosol water content. With decreased ammonium and sulfate, the available water content in aerosols decreases, adversely
influencing nitric acid partitioning to aerosols. Although reduced sulfate formation promotes more nitric acid partitioning,
thermodynamic calculations indicate the reduced water content, causing the observed decline in dissolved nitrate.

For NOx, reducing NOx emissions results in a lower $HNO_3$ formation, leading to a significant reduction in PM-nitrate. The
reduced acidity contribution from nitric acid alters the partition of ammonia, resulting in a decrease in ammonium. In contrast,
the slightly increased sulfate formation observed may be attributed to enhanced chemical processes under lower NOx
conditions. Reducing NOx emissions consumes less OH, a major pathway for $HNO_3$ formation during the daytime, as depicted
in Reaction 2 of Table 2. The increased availability of OH can enhance the oxidation of $SO_2$ to form sulfuric acid through
Reaction 1 in Table 2 (Derwent et al., 2009).

Regarding $NH_3$ emission reduction, $NH_3$ primarily acts as a base, influencing the dissolution of volatile acids such as $HNO_3$.
With nitrate having a higher molecular weight than ammonium, the most significant decrease in mass concentration is observed
for nitrate. Sulfuric acid, with negligible volatility, predominantly participates in the aerosol phase. The observed decrease in
sulfate is likely attributed to altered chemical processes influenced by $NH_3$, particularly the aqueous reactions. Further
exploration of the interplay between $NH_3$ reduction and sulfate will be discussed in more detail in Section 3.4.

With the reduction in $PM_{2.5}$ attributed to NOx or $NH_3$, the response of $PM_{2.5}$, sulfate, ammonium, and nitrate at various levels
of reduction in both NOx and $NH_3$ are shown in Fig. S7. For a given reduction in NOx (or $NH_3$), the trend of interested species
as a function of $NH_3$ (or NOx) is consistent, similar to the case discussed in Fig. 3. When both emissions are reduced, the
contour of $PM_{2.5}$ is concave upward, indicating that the concentration is lower than the linear combination of individual



influence on PM$_{2.5}$. A similar pattern happened to PM-nitrate and PM-ammonium, while PM-sulfate exhibits a relatively small and different change trend. The results suggest that the change in PM$_{2.5}$ concentration is mainly dominated by nitrate and ammonium, with sulfate having a minor effect. The deviation from the linear combination of individual contributions might be due to the variation in the partitioning between gas and aerosol phases under different acidity. For example, NH$_3$ might

have an increased portion in the gas phase as NOx is decreased, while HNO$_3$ would have a higher portion in the gas phase as NH$_3$ is reduced. The further reduction of the other species would reflect a lower portion reduction of PM-related species. Therefore, it can be inferred that as emission reduction reaches a certain level, the available nitric acid or ammonia is continuously reduced to very low levels, potentially leading to a decline in the efficiency of PM$_{2.5}$ emission reduction (Veratti et al., 2023).

**3.3.2 Sensitivity analysis**

The sensitivity evaluation for different emission species on PM-nitrate and PM$_{2.5}$ is shown in Fig. 4. In the case of PM-nitrate sensitivity, $S_{NOx,NO_3}$ increases as the emission ratio decreases and reaches a maximum value of 0.83 at the emission ratio around 0.4-0.8 (using control run as a base value). Subsequently, $S_{NOx,NO_3}$ gradually decreases as the emission ratio decreases. This transition in the $S_{NOx,NO_3}$ is likely due to the available quantity of NO$_2$ for HNO$_3$ formation via NO$_2$ + OH reaction. In addition

to being produced by chemical reactions, HNO$_3$ concentration is also affected by the transported HNO$_3$ from the domain boundaries. While the transported HNO$_3$ concentration is relatively low compared to HNO$_3$ produced by local NOx emissions in Taiwan, its proportion gradually increases as NOx emissions decrease. When the HNO$_3$ concentration produced by the local chemical reaction is comparable to the transported concentration, the sensitivity coefficient decreases. (Detailed mathematical verification is provided in Section S1).

In contrast, $S_{NH_3,NO_3}$ increases monotonically as the emission ratio decreases within the studied range, showcasting significant influence of local NH$_3$ emission on PM-nitrate quantity. In the studied environment context, a higher $S_{NOx,NO_3}$ than $S_{NH_3,NO_3}$ indicates that PM-nitrate is more sensitive to NOx emission in central Taiwan. The sensitivity on PM-nitrate in spatial distribution (Fig. S8a) shows that the major cities in western Taiwan and the southwest offshore are in a NOx-sensitive environment, while only the eastern region is biased toward NH$_3$-sensitive, likely due to relatively low NH$_3$ and NOx emissions.

As to the sensitivity on PM$_{2.5}$, the trend of NOx sensitivity ($S_{NOx,PM_{2.5}}$) is similar to $S_{NOx,NO_3}$, as a reduction in NOx emissions primarily leads to a decrease in nitrate, exerting a dominant influence on PM$_{2.5}$ concentration. $S_{NH_3,PM_{2.5}}$ and $S_{SO_2,PM_{2.5}}$ are relatively stable. $S_{SO_2,PM_{2.5}}$ gradually declines as the emission ratio decreases, while $S_{NH_3,PM_{2.5}}$ shows an increase first as the emission ratio > 0.8 and then decreases slightly. For the overall emission ratio range studied, $S_{NH_3,PM_{2.5}}$ is around 0.19±0.01 while $S_{NOx,NO_3}$ has a wider range from 0.05 to 0.24, and $S_{SO_2,PM_{2.5}}$ is 0.05±0.01. Under the studied condition (at the emission

ratio of 0.9 in Fig. 4b), $S_{SO_2,PM_{2.5}}$ (~ 0.05) is the lowest, $S_{NOx,PM_{2.5}}$ (~ 0.23) and $S_{NH_3,PM_{2.5}}$ (~ 0.2) are relatively higher, indicating that reducing NOx or NH$_3$ emissions results in a more significant reduction in PM$_{2.5}$ compared to reducing SO$_2$ emissions.



The sensitivity on $PM_{2.5}$ in spatial distribution (Fig. S8b) and the statistical data for each area (Table S7) show that $S_{NOx,PM_{2.5}}$ is greater than $S_{NH_3,PM_{2.5}}$ in each air pollution zone emphasizing the importance of NOx reduction in improving $PM_{2.5}$. However, in the northern, Chu-Miao, and central areas, $S_{NOx,PM_{2.5}}$ and $S_{NH_3,PM_{2.5}}$ are relatively close. These areas have some white shading, indicating neutrality, and suggest that the reduction of NOx and $NH_3$ emissions is equally important.

### 3.4 Emission affecting sulfate formation

### 3.4.1 Composite results in cloud

The observed decrease in sulfate levels with a lower $NH_3$ emission ratio (Figs. 3c and S7b) is likely attributed to the modified cloud pH, affecting aqueous phase sulfate production (Redington et al., 2009), as $NH_3$ emissions do not directly impact the gas-phase chemistry of $SO_2$. To preserve the critical features, the composite results from grid points containing clouds with significant $SO_2$ (i.e., a cloud water content $\geq 0.1$ g kg$^{-1}$ and $SO_2$ concentration $\geq 1$ ppbv when $NH_3$ is at 0.2x) are depicted in Figs. S9 and S10 for land and sea regions, respectively. This categorization takes into account variations in pollutant levels between these two regions, and the statistical summaries, including mean, 25th, and 75th percentile of cloud pH value and gaseous components, are provided in Table 4. While the mean pH values are higher over land than at sea, the majority of grids have a pH of 5, slightly below the average. Notably, grid points with lower pH values are predominantly characterized by $NH_3$ deficiency, especially at sea, where the concentrations of $NH_3$ are lower than those on land.

The changes in chemical substances for both land and sea show consistent trends with emission reduction, featuring with higher concentrations of sulfate at sea compared to on land. The pattern of sulfate formation in clouds is consistent with the average concentrations in the surface layer (Fig. S7b), increasing with NOx emission reduction and decreasing with $NH_3$ emission reduction. Conversely, the concentration change of $SO_2$ opposed that of sulfate due to the conservation of sulfur. A decrease in sulfate concentration implies that more sulfur remains as $SO_2$ in the atmosphere, indicating weaker oxidation reactions. Furthermore, the reduction of $NH_3$ also impacts primary oxidants involved in sulfur oxidation, namely $H_2O_2$ and $O_3$. Intriguingly, the changes in $H_2O_2$ and $O_3$ exhibit opposite trends in response to $NH_3$ emission reduction, suggesting a potential influence on their oxidation rates.

The aqueous oxidation pathways of $SO_2$ are strongly pH-dependent. The oxidation rate by $H_2O_2$ increases with pH < 3 and remains fairly constant at pH > 3 (Seinfeld et al., 2006). The other three reactions ($O_3$ and $O_2$ catalysis by Fe(III) and Mn(II)) are pH-dependent and increase with pH. Overall, $H_2O_2$ oxidation is usually a major process. However, with an increase in pH, the oxidation rates of $O_3$ and $O_2$ catalysis via Fe(III) or Mn(II) might surpass that of $H_2O_2$ if high enough Fe(III) or Mn(II) relative to $H_2O_2$ presents.

### 3.4.2 Case analysis of a single grid point

The condition of a grid point along the coast of Taichung (24.203° N, 120.5053° E, the second layer, ~ 68.5 m a.s.l.) at 8:00 am local time on 3$^{rd}$ December 2018 from CMAQ was selected. This grid point fulfills our desired criteria, featuring a cloud



water content of 0.376 g kg$^{-1}$ and an SO$_2$ concentration of 7.13 ppbv. Figure 5 shows the concentrations of SO$_2$, H$_2$O$_2$, O$_3$, and acidity (i.e., [H$^+$] in cloud water, calculated from CMAQ output data) at this grid point from CMAQ as a function of NH$_3$ and

NOx emission ratios. With NH$_3$ emission reduction, SO$_2$ concentration increases significantly, especially when the NH$_3$ emission ratio is below 0.4, while the concentration decreases slightly as the NOx emission decreases. The pattern for acidity mirrors that of SO$_2$, showing an increase as NH$_3$ emissions decrease and a smooth decrease as NOx emissions decrease. This suggests a possible strong correlation between SO$_2$ and acidity. As NOx emission decreases, the concentrations of both H$_2$O$_2$ and O$_3$ increase due to changes in gaseous chemical reactions that reduce the consumption of OH and O$_3$. When NH$_3$ emissions

decrease, O$_3$ increases and H$_2$O$_2$ decreases. The trend might indicate that efficient SO$_2$ oxidation via the O$_3$ reaction dominates at a high NH$_3$ emission ratio. With O$_3$ in excess of SO$_2$, SO$_2$ is completely reacted. As NH$_3$ emission is reduced to less than 0.6, the increased acidity significantly slows down SO$_2$ oxidation via the O$_3$ reaction, and the system switches to H$_2$O$_2$ oxidation.

Based on the observation trend, we can derive an initial condition of H$_2$O$_2$, O$_3$, SO$_2$, and sulfate for the box model under the

assumption of SO$_2$ oxidation purely by ozone for control run and H$_2$O$_2$ oxidation at NH$_3$ emission ratio = 0.2 (abbreviated as "NH3_02x run"). All other required parameters such as the concentrations of total nitric acid and dust are as assumed as the same as those in the control run. Therefore, the initial condition in the box model has H$_2$O$_2$ concentration from control run, the O$_3$ from NH3_02x run, and SO$_2$ concentration as SO$_2$ in NH3_02x run adding H$_2$O$_2$ difference between control run and NH3_02x run, the consumed SO$_2$ by the H$_2$O$_2$ oxidation reaction. The initial value of SO$_4$ is derived by subtracting the applied

SO$_2$ initial concentration from total S (i.e., PM-sulfate + SO$_2$ in control run). A summary of the initial conditions is provided in Table S3.

Figure 6 shows the comparison between box model results and corresponding CMAQ results. Considering the main aqueous phase reactions involving O$_3$ and H$_2$O$_2$, the box model findings demonstrate that as the initial total NH$_3$ concentration decreases, the pH and H$_2$O$_2$ also decrease, while SO$_2$ and O$_3$ increase. These trends are consistent with the pattern observed in

CMAQ but with some discrepancies. Specifically, in the box model, the concentration of SO$_2$ tends to be slightly higher at lower NH$_3$ emission ratios, whereas the concentrations of H$_2$O$_2$ and O$_3$ are lower than CMAQ results. Introducing additional oxidation reactions, i.e., the oxidation of tetravalent sulfur by O$_2$ with Fe(III) and Mn(II) catalysis in the system, bring the box model results closer to those of CMAQ. The box model demonstrates that higher NH$_3$ concentrations lead to higher pH values, resulting in O$_3$-dominated chemistry. The reduction of NH$_3$ emissions can increase environmental acidity, slow O3 oxidation

reactions, and gradually transition to an H$_2$O$_2$-dominated condition. However, in the studied environment, the concentration of H$_2$O$_2$ is lower than that of SO$_2$, resulting in residual SO$_2$ once H$_2$O$_2$ is depleted but causing a reduction in PM-sulfate.

### 3.5 Cost of emission reduction

The PM$_{2.5}$ reduction efficiency (Fig. 7a) based on the fitted trend of Fig. 3 increases as the emission ratio decreases in the three emission adjustment scenarios. In the studied emission condition (i.e., emission ratio = 1), the highest reduction efficiency is



for $NH_3$, followed by $SO_2$ and NOx with the lowest. The increasing trend with the reduction ratio suggests an expected higher $PM_{2.5}$ reduction efficiency as the emission control policies continue. However, as discussed in Section 3.3, reducing $SO_2$ emissions has the least significant improvement in $PM_{2.5}$. So, when the emission ratio is less than 0.8, its emission reduction efficiency is exceeded by reducing NOx emission. Hence, the available reduction capacity is the lowest for $SO_2$.

In the aspect of policy considerations, cost becomes a crucial factor. Fig. 7b illustrates the $PM_{2.5}$ reduction cost associated with these emission reduction experiments. The reduction of NOx emissions incurs the highest cost, amounting to approximately one billion dollars $yr^{-1}$ to achieve a 1 $\mu g$ $m^{-3}$ reduction in $PM_{2.5}$ concentration. In comparison, the cost of $SO_2$ emission reduction ranges from tens of millions to 100 million dollars $yr^{-1}$, while $NH_3$ emission reduction costs are around fifty million dollars $yr^{-1}$. Therefore, performing $NH_3$ and $SO_2$ emission reductions would be more cost-efficient in achieving a $PM_{2.5}$ reduction. However, cost evaluation involves uncertainties. Kaminski (2003) approach to estimating MAC of $SO_2$ focuses on power plants, discussing costs tied to emission control through alternative energy sources or equipment enhancements. On the other hand, Gu et al. (2021) employed the online Greenhouse Gas and Air Pollution Interactions and Synergies (GAINS) model to comprehensively assess the MAC of reducing NOx and $NH_3$ emissions across five continents and globally. Applying these methods to Taiwan may encounter uncertainties due to varying energy mixes, industrial structures, and environmental conditions in different regions. Such distinctions could diverge from prior study assumptions, affecting the cost-effectiveness of emission reduction strategies. Additional factors, including meteorological patterns and technological landscapes, may introduce uncertainties in cost estimations.

Furthermore, our study assumed a constant MAC value. In reality, MAC may vary as emissions decrease, usually becoming more expensive. Varied emission reduction approaches could result in substantial cost disparities, demanding careful consideration for regional applications. Therefore, a more comprehensive cost-benefit analysis, accounting for regional variations and potential changes in MAC with emission reduction, is crucial to devise effective and economically viable air pollution control strategies.

## 4 Conclusion

This study investigated the impacts of emission reduction on $PM_{2.5}$ and the secondary inorganic components (sulfate, nitrate, and ammonium) while assessing the effectiveness of emission reduction strategies in central Taiwan using the CMAQ model during December 2018. In our simulations, the mean $PM_{2.5}$ concentration in central Taiwan is 21.1 $\mu g$ $m^{-3}$, including 2.7 $\mu g$ $m^{-3}$ of sulfate, 6.3 $\mu g$ $m^{-3}$ of nitrate, 2.6 $\mu g$ $m^{-3}$ of ammonium and other species including organics. For sulfate, 43.7% comes from chemical processes, with 30.5 % from aqueous reactions and 13.2 % from gas-phase reactions.

In the evaluation of emission reduction, it was observed that the impact of $SO_2$ emission reduction on mitigating $PM_{2.5}$ was less significant compared to reductions in NOx and $NH_3$ emissions. This is attributed to the fact that $SO_2$ emission reduction primarily affects sulfate, which constitutes only 12 % of $PM_{2.5}$ in this studied case. On the other hand, the reduction of NOx or $NH_3$ emission substantially contributed to a significant decrease in nitrate and ammonium, effectively mitigating $PM_{2.5}$.



However, a non-linear effect exists between the emission reduction of NOx and NH$_3$, indicating that mitigating effects of these two emissions are not linearly additive. Through sensitivity analysis, the reduction of NOx or NH$_3$ emission is relatively close in northern Taiwan, Chu-Miao area, and central Taiwan. While the Yun-Chia-Nan area and Kao-Ping area prefer NOx emission reduction. Notably, NH$_3$ emission reduction affects sulfur chemical reactions in the aqueous phase through changing pH values in cloud droplets, switching the primary oxidant from ozone to H$_2$O$_2$, which is a limited agent in this studied case. The sensitivity of the oxidation reaction of S(IV) to S(VI) with respect to acidity was verified using the box model. The oxidations by O$_2$ catalyzed via Fe(III) and Mn(II) were also confirmed to have a significant contribution to the oxidation processes, as demonstrated using a box model. Other research also indicated the importance of metal ion-catalyzed sulfur oxidation reactions for the generation of sulfate. In winter, this pathway can contribute up to 19% of sulfate formation in China (Huang et al., 2014). Emission reduction strategies to combat PM$_{2.5}$ are crucial but entail considerable costs. Our comprehensive analysis reveals that, considering both efficiency and cost, reducing NH$_3$ emissions emerges as the most effective strategy for the studied Taiwan's environmental conditions. However, it is imperative to acknowledge that NH$_3$ emissions are mainly associated with industrial, agricultural, and livestock activities. Industrial ammonia manufacturing has greatly increased global food production and population in the past (Erisman et al., 2008), and green ammonia may also play a significant role in future carbon-free energy endeavors (Chehade et al., 2021; Kang et al., 2015). Therefore, exploring large-scale emission reduction strategies and carefully assessing potential issues, such as aerosol pH changes leading to increased acid rain (Liu et al., 2019), are vital areas for further research. Overall, this study provides valuable insights into the intricate interactions among air pollutants and their impacts on PM$_{2.5}$, highlighting the ongoing need for continued efforts to reduce emissions and improve air quality in Taiwan.

**Code availability**

The code is not publicly accessible, but readers can contact HM Hung (hmhung@ntu.edu.tw) for more information.

**Data availability**

The CMAQ model output and TW-MOENV observation data used in this study can be accessed online at https://doi.org/10.5281/zenodo.10623526.

**Author contributions**

PC Huang set up experiments, ran experiments, and prepared the draft. HM Hung supervised the project, including data discussion and manuscript editing. HC Lai provided the control run of WRF and CMAQ model. CCK Chou provided IC analysis of PM$_{2.5}$.



**Competing interests**

The authors declare that they have no conflict of interest.

**Acknowledgments**

This study is supported by National Science and Technology Council (NSTC), Taiwan under grants of 111-2111-M-002-009
and 112-2111-M-002-014. We acknowledge the valuable insights from Dr. Ruijun Dang at Harvard and Prof. Jen-Ping Chen
at National Taiwan University for the sensitivity analysis discussion.

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





**Tables**

**Table 1: Experimental design.**

| Experiments | Descriptions |
|---|---|
| **Control run** | Use mechanism cb6r3ae6aq |
| **NoAqChem run** | Turn off sulfur aqueous phase oxidation reactions |
| **NoChem run** | Turn off all chemistry reactions |
| **ER1 runs** | $SO_2$, NOx, and $NH_3$ emissions reduced by ratios of 0.8, 0.6, 0.4, and 0.2 separately |
| **ER2 runs** | Both NOx, and $NH_3$ emissions reduced by ratios of 0.8, 0.6, 0.4, and 0.2 (only 12/1~12/14) |
| **EI runs** | $SO_2$, NOx, and $NH_3$ emissions increased by ratios of 1.5 and 2.0 separately |





**Table 2: The applied chemical reactions related to the formation of sulfate, nitrate, and ammonium.**

|  |  | Reaction |
| --- | --- | --- |
| **Gas phase** | 1. | $SO_{2(g)} + OH_{(g)} \rightarrow HSO_{3(g)} \rightarrow\rightarrow H_2SO_{4(g)}$ |
|  | 2. | $NO_{2(g)} + OH_{(g)} \rightarrow HNO_{3(g)}$ |
|  | 3. | $N_2O_{5(g)} + H_2O \rightarrow 2HNO_{3(g)}$ |
| **Aqueous phase** | 4. | $H_2SO_{4(aq)} \rightarrow SO_{4(aq)}^{2-} + 2H^+$ |
|  | 5. | $S(IV)_{(aq)} + Oxidants_{(aq)} \rightarrow SO_4^{2-} + 2H^+$ |
|  |  | * S(IV): $H_2SO_3$, $HSO_3^-$, $SO_3^{2-}$ |
|  |  | * Oxidants: $O_3$, $H_2O_2$, MHP, PAA, $O_2$ |
|  | 6. | $NH_{3(g)} \leftrightarrow NH_{3(aq)} \leftrightarrow NH_{4(aq)}^+ + OH^-$ |
|  | 7. | $HNO_{3(g)} \leftrightarrow HNO_{3(aq)} \leftrightarrow NO_{3(aq)}^- + H^+$ |
|  | 8. | $H^+ + OH^- \rightarrow H_2O$ |






**Table 3: Statistic of wind and PM$_{2.5}$ of MOENV observation and model simulation for four stations.**

|  | Tamsui | Shalu | Taixi | Qianzhen | [a]Criteria |
|---|---|---|---|---|---|
| **Wind speed (m/s)** | | | | | |
| Mean value of MOENV | 1.99 | 4.52 | 7.52 | 2.07 | |
| Mean value of WRF | 3.25 | 4.73 | 4.85 | 2.33 | |
| Correlation coefficient | 0.46 | 0.85 | 0.69 | 0.42 | |
| Mean bias error | 1.22 | 0.21 | -2.70 | 0.25 | $\leq \pm 0.5$ |
| Mean absolute error | 1.56 | 1.11 | 3.00 | 0.77 | $\leq 2.0$ |
| **PM$_{2.5}$ concentration (µg m$^{-3}$)** | | | | | |
| Mean value of MOENV | 10.75 | 16.09 | 20.93 | 31.72 | |
| Mean value of CMAQ | 12.12 | 19.70 | 15.97 | 40.56 | |
| Correlation coefficient | 0.59 | 0.70 | 0.71 | 0.42 | |
| Mean bias error | 1.42 | 3.72 | -4.79 | 8.88 | |
| Mean absolute error | 8.02 | 9.36 | 10.37 | 15.02 | |
| Mean fractional bias | -0.32 | 0.21 | -0.48 | 0.19 | $\leq \pm 0.6$ |
| Mean fractional error | 0.71 | 0.56 | 0.66 | 0.40 | $\leq 0.75$ |

[a]The crieteria are suggested by Hu et al. (2016).

$$\text{Correlation coefficient} = \frac{\sum_{i=1}^{n}(m_i - \bar{m})(o_i - \bar{o})}{\sqrt{\sum_{i=1}^{n}(m_i - \bar{m})^2}\sqrt{\sum_{i=1}^{n}(o_i - \bar{o})^2}}$$

$\quad \text{Mean bias error} = \overline{(m_i - o_i)}$

$\text{Mean absolute error} = \overline{|(m_i - o_i)|}$

$\text{Mean fractional bias} = 2 \times \overline{\left(\frac{m_i - o_i}{m_i + o_i}\right)}$

$\text{Mean fractional error} = 2 \times \overline{\left|\left(\frac{m_i - o_i}{m_i + o_i}\right)\right|}$

where $m_i$ and $o_i$ are the wind speed or concentrations of model and observation at time i, respectively, and $\bar{m}$ and $\bar{o}$ are the

average values over December 2018.





**Table 4: Correlation coefficients of PM$_{2.5}$, sulfate, nitrate, ammonium, and meteorological parameters (WS and QC) of control run for central Taiwan.**

|  | PM$_{2.5}$ | SO$_4^{2-}$ | NO$_3^-$ | NH$_4^+$ | dSO$_4^{2-}$ |
|---|---|---|---|---|---|
| **SO$_4^{2-}$** | 0.65 | - | - | - | 0.73 |
| **NO$_3^-$** | 0.97 | 0.61 | - | - | - |
| **NH$_4^+$** | 0.95 | 0.77 | 0.98 | - | - |
| **WS** | -0.54 | -0.34 | -0.51 | -0.52 | - |
| **QC** | 0.05 | 0.43 | 0.07 | 0.18 | 0.65 |

WS is the single-point wind speed of the surface layer in Shalu.

QC is the average cloud water within the planetary boundary layer in central Taiwan.

dSO$_4^{2-}$ is the sulfate difference concentration (control run – NoAqChem run).





**Table 5: Statistics of pH in cloud droplets and the concentration of gaseous components in both land (32130 grid points) and sea (122316 grid points) regions.**

|  | Land |  |  | Sea |  |  |
|---|---|---|---|---|---|---|
| **Variables** | **Mean** | **Q1** | **Q3** | **Mean** | **Q1** | **Q3** |
| **pH** | 5.15 | 5.00 | 5.00 | 5.01 | 5.00 | 5.00 |
| **$SO_2$ (ppbv)** | 1.65 | 1.01 | 1.96 | 1.68 | 1.16 | 1.90 |
| **$NH_3$ (ppbv)** | 2.10 | 0.03 | 0.84 | 0.42 | 0.02 | 0.16 |
| **$HNO_3$ (ppbv)** | 0.37 | 0.11 | 0.51 | 1.42 | 0.61 | 1.72 |
| **$H_2O_2$ (ppbv)** | 0.06 | 0.003 | 0.06 | 0.05 | 0.002 | 0.03 |
| **$O_3$ (ppbv)** | 44.6 | 38.9 | 51.3 | 48.2 | 43.2 | 53.6 |

Mean: Arithmetic mean; Q1: 25th percentile; Q3: 75th percentile.





**555 Figures**

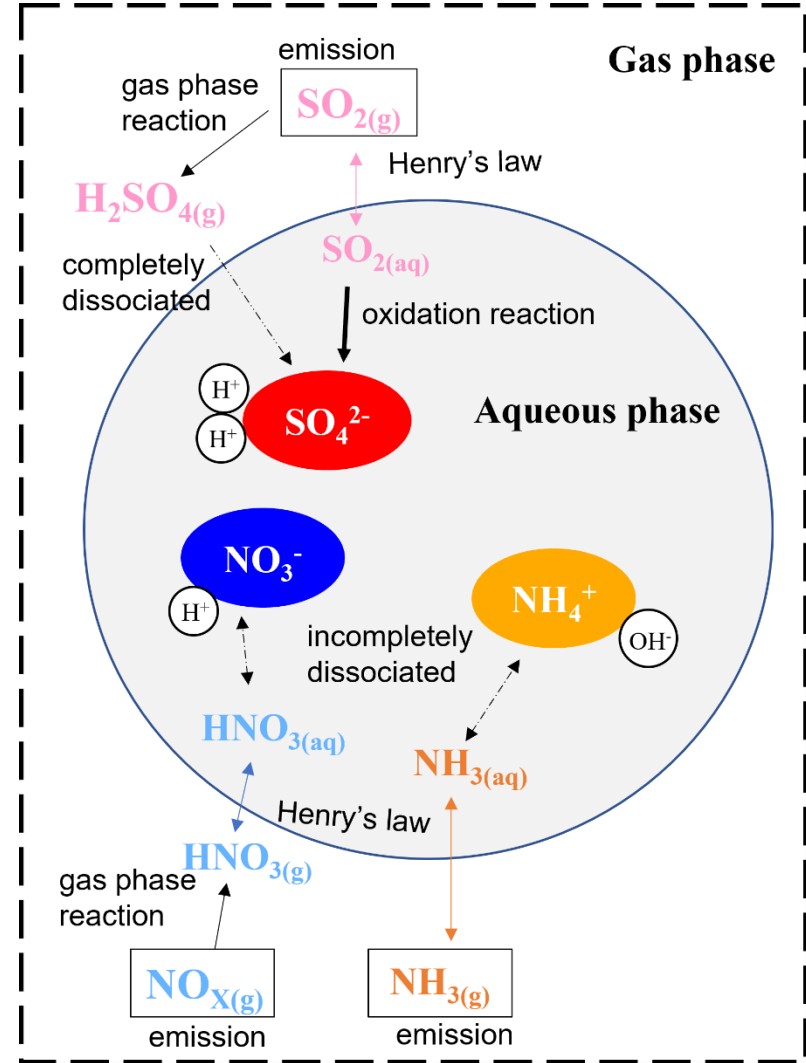

**Figure 1: Schematic diagram of chemical interaction for sulfate-nitrate-ammonium formation.**



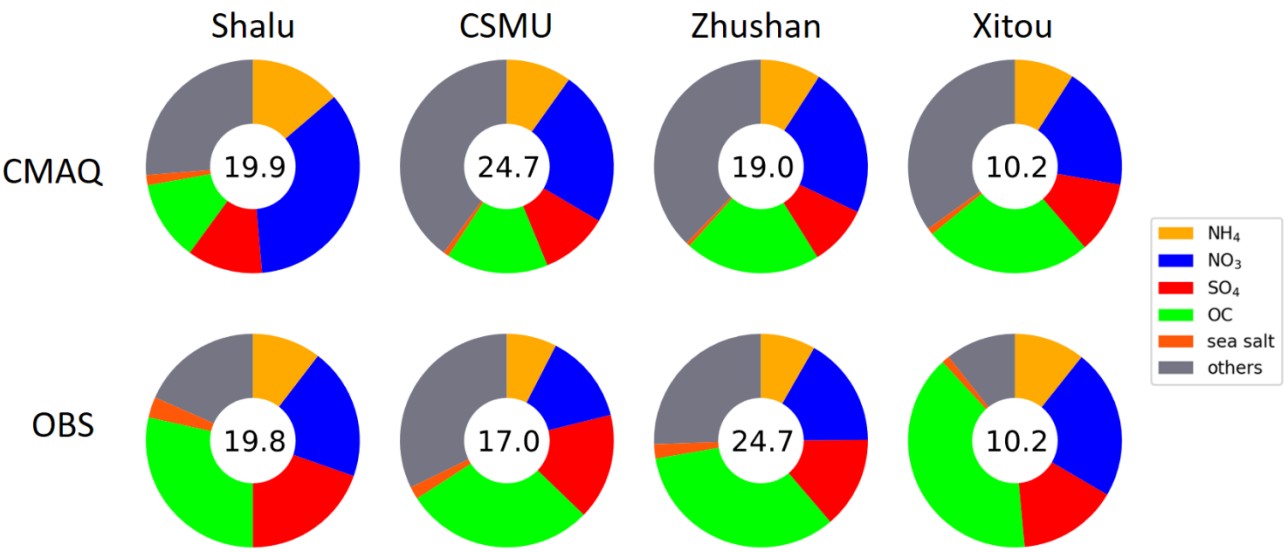

**Figure 2: The comparison of PM$_{2.5}$ components (the PM$_{2.5}$ concentration is indicated as a number inside the circle, μg m$^{-3}$) between intensive observation data and CMAQ surface layer data for four sites. The individual composition is shown in the legend, with colors arranged in a clockwise direction starting from the top. Conditions: mean values from 1-21 December 2018 for each station (OBS) or grid point (CMAQ).**






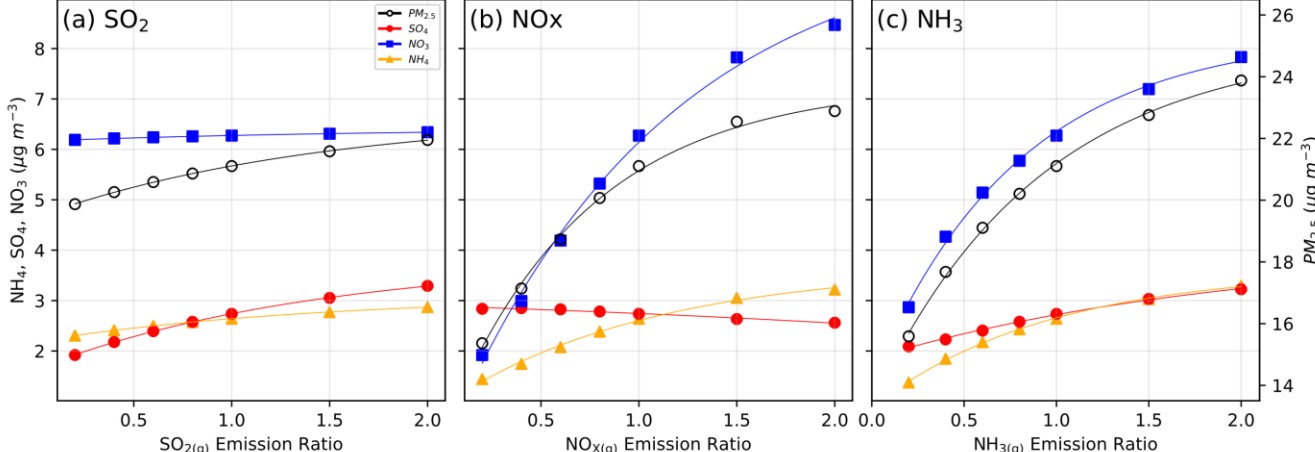

**Figure 3: The response of PM$_{2.5}$ and major secondary inorganic components (sulfate, nitrate, and ammonium) to the**
**emission ratio of (a) SO$_2$, (b) NOx, and (c) NH$_3$. Lines are applied to fit the data using an exponential function.**
**Conditions: averaged data of central Taiwan from 1-31 December 2018 for the surface layer.**

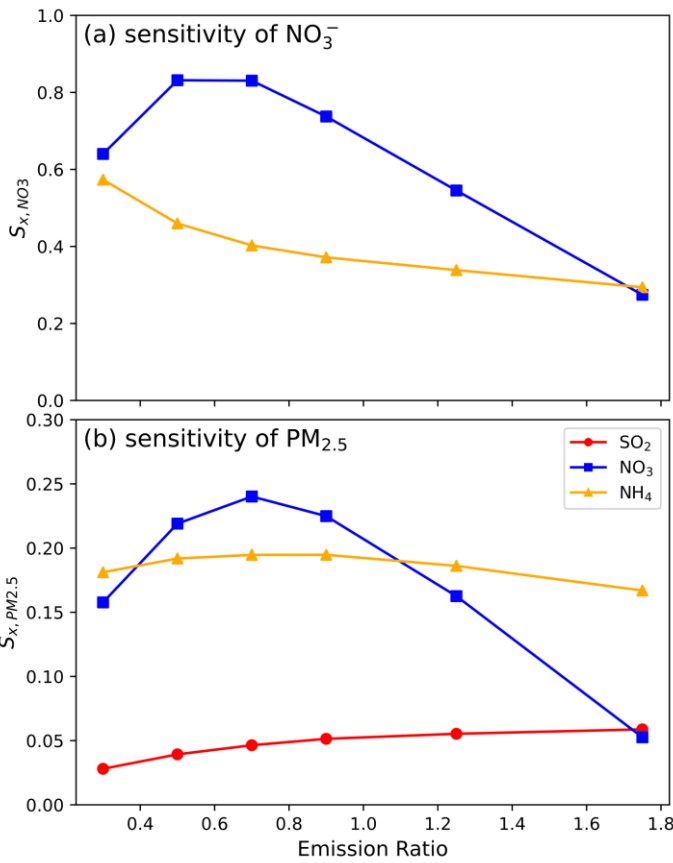

**Figure 4: (a) Nitrate sensitivity coefficient of NOx ($S_{NOx,NO_3}$) and NH₃ ($S_{NH_3,NO_3}$) and (b) PM₂.₅ sensitivity coefficient of SO₂ ($S_{SO_2,PM_{2.5}}$), NOx ($S_{NOx,PM_{2.5}}$) and NH₃ ($S_{NH_3,PM_{2.5}}$) as a function of emission ratio. Points are calculated using the first-order difference, with the x-axis representing the mean values of the two points involved in the differencing process. Conditions: averaged data of central Taiwan from 1-31 December 2018 for the surface layer.**








**Figure 5:** (a) SO₂, (b) hydrogen ion, (c) H₂O₂, and (d) ozone concentration as a function of NOx (x-axis) and NH₃ (y-axis) emission ratios for a single grid point. Conditions: 3 December 2018 08:00 LT, location of (24.2° N, 120.5° E), with an altitude of 68.5m above sea level.

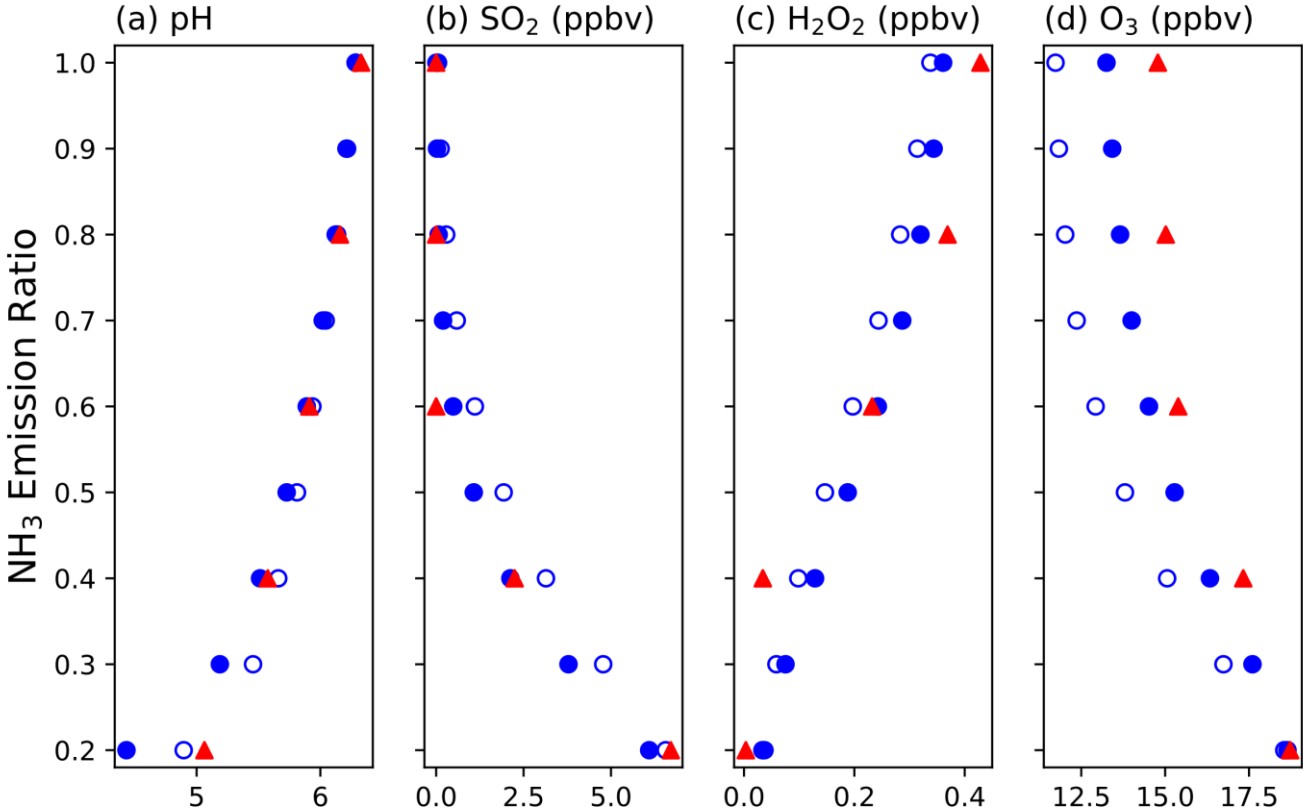

**Figure 6: The comparison between CMAQ and box model for (a) pH and the concentration of (b) SO$_2$, (c) H$_2$O$_2$, and (d) ozone as a function of NH$_3$ emission reduction ratio. Red triangle points: CMAQ model results; blue open circles: box model with H$_2$O$_2$ and O$_3$ reactions; blue solid circles: box model with H$_2$O$_2$, O$_3$, and O$_2$ catalyzed by Fe(III) and Mn(II) reactions.**





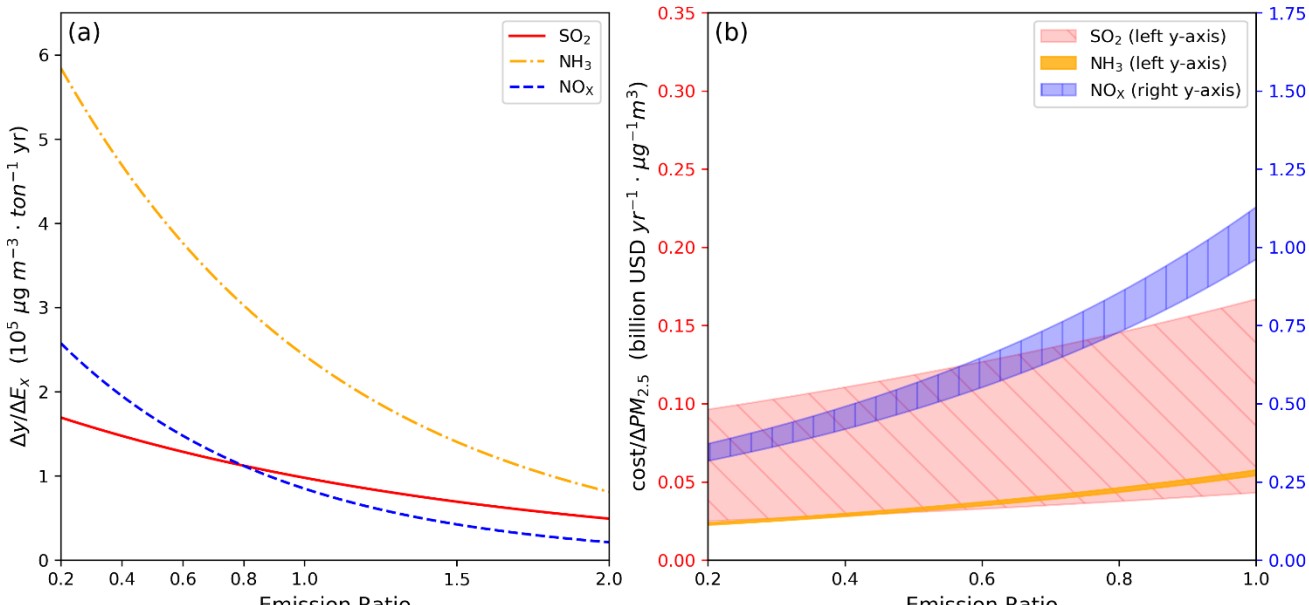

**Figure 7: (a) PM$_{2.5}$ reduction efficiency and (b) reduction cost as a function of emission ratio for SO$_2$, NH$_3$, and NOx.**