# Peer review of "Assessing the Effectiveness of SO2, NOx, and NH3 Emission Reductions in Mitigating Winter PM2.5 in Taiwan Using CMAQ Model"

_EGUsphere, 2024_

## Author Comment (AC1)

**We would like to thank the anonymous reviewer for the comments that significantly improved the clarity and readability of the manuscript. Our point-by-point responses are found below in blue ink. The revised content is highlighted in yellow.**

This study assessed the effectiveness of reducing NH3, NOx, and SO2 emissions on PM2.5 in December 2018 by using the CMAQ model. In general, the method is well recognized, and the study is logically designed. A few modifications and clarifications are needed.

1. The title can be modified as 'mitigating winter PM2.5 in Taiwan' to be more accurate.

A: Thanks for the reviewer's comment. The title was revised to "Assessing the Effectiveness of $SO_2$, NOx, and $NH_3$ Emission Reductions in Mitigating Winter $PM_{2.5}$ in Taiwan Using CMAQ Model".

2. Equation 3: I don't understand how the log calculation appeared here.

A: Based on the chain rule of logarithmic differentiation, $\frac{d(\log y)}{dx} = \frac{1}{y} \cdot \frac{dy}{dx}$. In equation 3, the calculation processes was simplified as $\frac{E_X}{Y}\frac{\Delta Y}{\Delta E_X} = \frac{\Delta \log(Y)}{\Delta \log(E_X)}$ (In our study, we use $\Delta$ instead of $d$ because our current data are discrete rather than continuous.) To reduce the confusion, the equation in the content was revised as

$$S_{X,Y} = \frac{E_X}{Y}\frac{dY}{dE_X} = \frac{d\log(Y)}{d\log(E_X)} \approx \frac{\Delta \log(Y)}{\Delta \log(E_X)}$$

3. Model performance: For evaluation model performance on meteorology and air quality, there are certain criteria and statistical matrix to evaluate. The model performance can be accepted when compared to these criteria. A reference can be: Atmos. Chem. Phys., 16, 10333–10350, 2016.

A: We appreciate the reviewer's suggestion. The mean fractional bias and mean fractional errors were provided for further criteria comparison. The information is incorporated in Table 3, while the first paragraph in section 3.1.1 was revised to address this analysis as follows: "The comparison between WRF model results and TW-MOENV observations …, mean bias errors, mean absolute error, mean fractional bias, and mean fractional errors. …. The mean bias error at Shalu and Qianzhen meets the criteria suggested by Hu et al. (2016), while the mean absolute error at Tamsui, Shalu, and Qianzhen meets the criteria. At Taixi, the model tends to be underestimated, resulting in a higher mean absolute error. Overall, these findings demonstrate satisfactory model performance." In addition, the following sentence is added to section 3.1.2. (Lines 204-205) "The correlation coefficients for $PM_{2.5}$ concentration range from 0.42 to 0.71, and the mean fractional bias and mean fractional error for $PM_{2.5}$ are within the acceptable criteria (Table 3), affirming the model's reliability (Fig. S2)."

4. Some more detailed discussion on performance on the components should be provided, such as time series plots of obs vs. pre sulfate, nitrate and ammonium, as these are the core of the study.

A: We appreciate the reviewer's suggestion. Figure S6 is added to show time series plots of sulfate, nitrate, and ammonium with the following content added to section 3.1.2 (Lines 221-227). "The correlation coefficients of $PM_{2.5}$ between observation and model at Shalu and CSMU are 0.76 and 0.65, respectively, demonstrating consistency of model results for concentration and change trend at these two stations (Fig. S5). However, the correlation between observation and model at Zhushan and Xitou is poor, likely due to the influence of the complex topography at these two places. Further analysis in Fig. S6 presents the trends and correlation coefficients for PM-sulfate, PM-nitrate, and PM-ammonium across the four stations. The data reveal a slight underestimation trend for PM-sulfate, particularly at Shalu and Zhushan. The simulation for PM-ammonium appears reasonably accurate, whereas PM-nitrate shows a tendency for overestimation. …"

5. The results in Fig. S7 and S9 are averages from 1-14 December. Why?

A: In Fig. 2, the $PM_{2.5}$ concentration trend throughout December exhibits two cycles of a high pollution period followed by a clean period. The results using the first cycle are consistent with the changing trend of the monthly data for reducing single-component emissions, as shown in Fig. R1. Therefore, to conserve computational resources, the "ER2 runs" experiments are only performed for the first half of the month.

[Figure]

Figure R1: The response of $PM_{2.5}$ and major secondary inorganic components (sulfate, nitrate, and ammonium) to the emission ratio of (a) NOx and (b) $NH_3$. Solid lines are the average data of December 2018 for the surface layer of central Taiwan, and the dashed

lines are the average data from 1ˢᵗ to 14ᵗʰ December 2018 for the surface layer of central Taiwan.

6. It will be more interesting to see whether the 'effectiveness' differs during the 'high pollution' period (such as beginning of December and middle December) and during relatively clean period.

A: We appreciate the reviewer's suggestion. Figures S12 and S13 were added to show the effectiveness of emission reduction during the high pollution period and relatively clean period. The following sentences were added to section 3.5 (Lines 373-379). "... Additionally, the $PM_{2.5}$ reduction efficiency during relatively clean period and high pollution period is presented in Figs. S12a and S13a, respectively. During the clean period (6ᵗʰ to 12ᵗʰ December), $NH_3$ reduction maintains the highest efficiency, followed by $SO_2$ and NOx. However, during the high pollution period (16ᵗʰ to 22ⁿᵈ December), $NH_3$ reduction still has the highest efficiency, but NOx is higher than $SO_2$. This indicates that during high pollution periods, reducing $SO_2$ emission has a limited effect on the total amount of $PM_{2.5}$ concentration, and continued reducing $SO_2$ emission does not improve efficiency. The average results of these two different conditions explain the crossover pattern observed for $SO_2$ and NOx emission reduction in Fig. 7a."

[Figure]

**Figure S12: (a) $PM_{2.5}$ reduction efficiency and (b) reduction cost as a function of emission ratio for $SO_2$, $NH_3$, and NOx during the clean period of 6ᵗʰ-12ᵗʰ December 2018.**

[Figure]

**Figure S13: (a) $PM_{2.5}$ reduction efficiency and (b) reduction cost as a function of emission ratio for $SO_2$, $NH_3$, and NOx during the high pollution period of 16ᵗʰ-22ᵗʰ December 2018.**

---

## Author Comment (AC2)

**We would like to thank the anonymous reviewer for the comments that significantly improved the clarity and readability of the manuscript. Our point-by-point responses are found below in blue ink. The revised content is highlighted in yellow.**

This study is of value in that it provides $PM_{2.5}$ species measurements in Taiwan and assessed the emission control effects. However, the paper is not presented in a professional way. Many places are using non-scientific expressions in the field of atmospheric chemistry. Therefore, the whole paper needs to be substantially improved before it can be published on ACP. The problematic wording includes but not limited to –

1. Abstract line 1: "when particulate matter ($PM_{2.5}$) levels …" should be "when fine particulate matter ($PM_{2.5}$) levels …"
2. Line 13 please revise "In contrast, local NOx …"
3. Line 35, it is misleading to say "such as sulfate, nitrate, and ammonium" after "gas-phase precursors".
4. 2.3.1 section title: "sulfate sources" sounds better than "sulfate contribution".
5. 3.2 section title "sulfate formation pathways".

A: Thanks for the reviewer's comment. We have reviewed the content to correct the wording and present it in a more professional way. Some examples are provided as follows:

Abstract line 1: "when fine particulate matter ($PM_{2.5}$) levels …"

Lines 13-14: "In contrast, nitrate and ammonium are predominantly influenced by local NOx and $NH_3$ emissions. Reducing $SO_2$ emissions decreases sulfate levels, which in turn affects $NH_3$ partitioning and results in lower ammonium concentrations."

Lines 34-36: "PM can enter the atmosphere through direct emissions of primary aerosols, such as black carbon, sea salt, dust, and certain organic substances. Alternatively, PM can be formed via chemical reactions of gas-phase precursors, creating secondary aerosols such as sulfate ($SO_4^{2-}$), nitrate ($NO_3^-$), and ammonium ($NH_4^+$) (Seinfeld et al., 2006)."

2.3.1 section title: "Sulfate sources".

3.2 section title: "Sulfate formation pathways".

Lines 21-23: "Nevertheless, the costs of emission reduction vary due to differences in methodology and regional emission sources."

Lines 98-99: "Additionally, intensive observation data using filter sampling were obtained from Shalu…"

Lines 208-210: "To assess regional distribution, we used area average concentration and partitioning of $PM_{2.5}$, based on TW-MOENV's pollutant zone classification (Fig. S3b),

focusing on areas with elevation less than 200 m above sea level (a.s.l.) to avoid complexities in terrain"

Lines 221-222: "The correlation coefficients of $PM_{2.5}$ between observation and model at Shalu and CSMU are 0.76 and 0.65, respectively, demonstrating consistency of model results for concentration and change trend at these two stations (Fig. S5)."

Line 345: "This suggests a strong correlation between $SO_2$ and acidity, likely due to a common influencing factor, $NH_3$."

---

## Author Response (AR2)

**We would like to thank the anonymous reviewer for the comments that significantly improved the clarity and readability of the manuscript. Our point-by-point responses are found below in blue ink. The revised content is highlighted in yellow.**

Editor: I agree with referee #1's further comments that the abstract can be enhanced. Please consider adding some quantitative statements (with specific values) to make your conclusions in the abstract clearer.

Referee #1: This work is of value for effective PM2.5 control in Taiwan. I suggest the authors revise the abstract to make the conclusions clearer and sharper. For example, could the authors spell out what "changes in HNO3 and NH3 partitioning" in lines 15-16?

A: Thanks for the reviewer and editor's comments. The abstract has been revised to include some quantitative statements and to clarify "changes in $HNO_3$ and $NH_3$ partitioning" as follows:

"Taiwan experiences higher air pollution in winter when fine particulate matter ($PM_{2.5}$) levels frequently surpass national standards. ... In contrast, nitrate and ammonium are predominantly influenced by local NOx and $NH_3$ emissions. Reducing $SO_2$ emissions decreases sulfate levels, which in turn leads to more $NH_3$ remaining in the gas phase, resulting in lower ammonium concentrations. Similarly, reducing NOx emissions lowers $HNO_3$ formation, impacting nitrate and ammonium concentrations by decreasing the available $HNO_3$ and leaving more $NH_3$ in the gas phase. A significant finding is that reducing $NH_3$ emissions decreases not only ammonium and nitrate but also sulfate by altering cloud droplet pH and $SO_2$ oxidation processes. While the impact of $SO_2$ reduction on $PM_{2.5}$ is less than that of NOx and $NH_3$, it emphasizes the complexity of regional sensitivities. Most of western Taiwan is NOx-sensitive, so reducing NOx emissions has a more substantial impact on lowering $PM_{2.5}$ levels. However, given the higher mass emissions of NOx than $NH_3$ in Taiwan, $NH_3$ has a more significant consequence in mitigating $PM_{2.5}$ per unit mass emission reduction (i.e., $2.43 \times 10^{-5}$, and $0.85 \times 10^{-5}$ $\mu g\ m^{-3}$ $ton^{-1}$ yr for $NH_3$, and NOx, respectively, under current emission reduction). The cost-effectiveness analysis suggests that $NH_3$ reduction outperforms $SO_2$ and NOx reduction (i.e., 0.06, 0.1, and 1 billion USD $yr^{-1}$ $\mu g^{-1} m^3$ for $NH_3$, $SO_2$, and NOx, respectively, under current emission reduction). Nevertheless, the costs of emission reduction vary due to differences in methodology and regional emission sources. Overall, this study considers both efficiency and costs, highlighting $NH_3$ emissions reduction as a promising strategy for $PM_{2.5}$ mitigation in the studied Taiwan's environment.